# Evaluation of the Antioxidant Activity of *Nigella sativa* L. and *Allium ursinum* Extracts in a Cellular Model of Doxorubicin-Induced Cardiotoxicity

**DOI:** 10.3390/molecules25225259

**Published:** 2020-11-11

**Authors:** Raluca Maria Pop, Ioana Corina Bocsan, Anca Dana Buzoianu, Veronica Sanda Chedea, Sonia Ancuța Socaci, Michela Pecoraro, Ada Popolo

**Affiliations:** 1Department of Pharmacology, Toxicology and Clinical Pharmacology, “Iuliu Hatieganu” University of Medicine and Pharmacy, Victor Babes, No 8, 400012 Cluj-Napoca, Romania; corinabocsan@yahoo.com (I.C.B.); ancabuzoianu@yahoo.com (A.D.B.); 2Research Station for Viticulture and Enology Blaj (SCDVV Blaj), 515400 Blaj, Romania; chedeaveronica@yahoo.com; 3Department of Food Science, University of Agricultural Sciences and Veterinary Medicine of Cluj-Napoca, Calea Manaștur 3–5, 400372 Cluj-Napoca, Romania; sonia.socaci@usamvcluj.ro; 4Department of Pharmacy, University of Salerno, via Giovanni Paolo II 132, 84084 Fisciano (SA), Italy; mipecoraro@unisa.it

**Keywords:** black cumin, wild garlic, cardiotoxicity, antioxidant activity, phenolic compounds

## Abstract

Natural products black cumin—*Nigella sativa* (*N. sativa*) and wild garlic—*Allium ursinum* (AU) are known for their potential role in reducing cardiovascular risk factors, including antracycline chemotherapy. Therefore, this study investigates the effect of *N. sativa* and AU water and methanolic extracts in a cellular model of doxorubicin (doxo)-induced cardiotoxicity. The extracts were characterized using Ultraviolet-visible (UV-VIS) spectroscopy, Fourier-transform infrared (FT-IR) spectroscopy, Liquid Chromatography coupled with Mass Spectrometry (LC-MS) and Gas Chromatography coupled with Mass Spectrometry (GC-MS) techniques. Antioxidant activity was evaluated on H9c2 cells. Cytosolic and mitochondrial reactive oxygen species (ROS) release was evaluated using 2′,7′-dichlorofluorescin-diacetate (DHCF-DA) and mitochondria-targeted superoxide indicator (MitoSOX red), respectively. Mitochondrial membrane depolarization was evaluated by flow cytometry. LC-MS analysis identified 12 and 10 phenolic compounds in NSS and AU extracts, respectively, with flavonols as predominant compounds. FT-IR analysis identified the presence of carbohydrates, amino acids and lipids in both plants. GC-MS identified the sulfur compounds in the AU water extract. *N. sativa* seeds (NSS) methanolic extract had the highest antioxidant activity reducing both intracellular and mitochondrial ROS release. All extracts (excepting AU methanolic extract) preserved H9c2 cells viability. None of the investigated plants affected the mitochondrial membrane depolarization. *N. sativa* and AU are important sources of bioactive compounds with increased antioxidant activities, requiring different extraction solvents to obtain the pharmacological effects.

## 1. Introduction

Globally, cardiovascular diseases (CVDs) are continuing to be the leading cause of death. According to the World Health Organization (WHO) data. Heart attack, stroke, and heart failure cause around 17.9 million deaths each year [1]. Among the most common risk factors with importance in CVDs etiology and development, are the existing comorbidities such as hypertension, hyperlipidemia, obesity, diabetes mellitus, or others such as genetic factors, age and gender, pharmacological treatments, dietary factors, tobacco use, excessive alcohol use and physical inactivity [2,3]. Antracycline chemotherapy is a well-known high-risk factor for cardiomyopathy since the development of heart failure may occur in up to 65% of doxorubicin (doxo -treated patients [4,5]. Current treatments used in the prevention of antracycline cardiotoxicity are few, and most of the time limited to the renin angiotensin system blockade, the iron chelator dexrazoxane and beta blockers [3]. However, the existing ways used to reduce antracyclyne cardiotoxicity, primary or secondary, by the use of classical treatments or by counselling the patients to dietary and lifestyle changes, still needs improvements [6].

Unfortunately, the existing synthetic therapeutic drugs available for the treatment of different cardiovascular disorders occur with multiple sides effects, and sometimes their use is ineffective for numerous patients [3,7]. Moreover, the interaction of cardiovascular medication with other drugs sometimes limits their use [8]. In this context, the uses of other alternatives like nutraceuticals (herbals, nutrients, phytochemicals, functional foods and others) have had an increased and continuous growth over the time, best observed in their market sales [9]. Accordingly, in 2018 the nutraceuticals market in the USA accounted for US$73,986.0 M. Moreover, between 2019–2027 it is expected to grow to US$138,047.1 M [10]. This aspect should also be closely monitored since the use of medicinal plants concomitantly with other medications can induce side effects, toxicity or possible herb-drug interactions [11]. Thus, both health benefits and possible side effects of nutraceuticals should be exhaustively tested before their use.

Among the most common nutraceuticals, herbals were used for centuries in traditional medicine to treat or to prevent different types of diseases. With respect to CVDs, black cumin (*Nigella sativa*—*N. sativa*) and wild garlic (*Allium ursinum*—AU) have been long experienced [12,13].

*N. sativa,* through its rich composition in bioactive compounds (thymoquinone, thymol, carvacrol, p-cymene, y-terpinene, a-thujene, and phenolic acids) possesses strong activity against multiple cardiovascular risk factors [14]. Thus, administration of *N. sativa* in various experimental settings has proven to improve dyslipidemia parameters through a decrease in total triglycerides, total cholesterol, and LDL-cholesterol levels, and an increase of HDL-cholesterol levels. Its administration proved to have an important antihypertensive effect, mainly demonstrated through its hypoglycemic [15], hypolipidemic [16], antioxidant effect [17], cardiac depressant effects [18]; calcium channel blockade effect [19] and its diuretic effects [20]. It is also beneficial in coronary heart disease management [21] helping in the prevention of atherosclerotic plaque formation [22], improving contractile and vascular heart activities [23] or increasing maximal oxygen consumption [24].

Regarding AU, it was demonstrated that its rich composition in sulfur-containing compounds (alliin, allyl methyl thiosulfonate, γ-l-glutamyl-S-alkyl-l-cysteine) [25], volatile oils (disulfides, trisulfides, tetrasulfides, and the nonsulfur compounds) [26] polyphenols (phenolic acids, flavonol glycosides derivatives) [27,28], steroidal glycosides [26] dietary fiber and microelements [27] are responsible for its pharmacological properties.

Evidence from various studies indicates that its protective effect against CVD risk factors are related mainly to its lowering lipid profile effect [25], reducing effect on blood pressure levels [29], antiatherogenic effect [30], antiplatelet activity [31,32], increasing fibrinolytic and antioxidant activity [25].

Natural products seem to be able to limit doxo-induced cardiotoxicity characterized by progressive myocardial stress, cardiomyopathy and heart failure [33]. The reasons for doxo-induced cardiotoxicity are multifactorial, but oxidative stress and mitochondrial dysfunction play a central role [5,34]. So, the association of doxo with the antioxidants represents a good perspective in reducing its side effects, and the identification of synthetic or natural compounds capable of reducing the doxo-induced cardiotoxicity is of great interest. Therefore, this study was aimed to investigate the effect of *N. sativa* seeds (NSS) and AU in a cellular model of doxo-induced cardiotoxicity.

## 2. Materials and Methods

### 2.1. Reagents and Standards

Acetonitrile, methanol and acetic acid used for HPLC-MS analysis were purchased from Merck (Darmstadt, Germany). All other chemicals used for sample extractions were of analytical grade and were purchased from Sigma-Aldrich (Steinheim, Germany). Water was freshly prepared using the Milli-Q purification system for water (Millipore, Billerica, MA, USA). Doxo was purchased from Baxter manufacturing Spa (Officina di Sesto Fiorentino, Florence, Italy). Embryonic rat heart cardiomyocyte-derived cell line H9c2 were purchased from the American Tissue Culture Collection (Manassas, VA, USA).

### 2.2. Plant Material

Dried AU leaves and NSS were purchased from PlantExtrakt, Cluj-Napoca, Romania where they were sold as food supplements. The botanical identification was done by the specialists from the quality control laboratory of the PlantExtrakt Company (Rădaia, Cluj, Romania).

### 2.3. Extraction Method

Dried leaves of AU (15 g) and dried NSS (15 g) were grounded to a fine powder and further sonicated in 300 mL of water or 300 mL of methanol, respectively. After 30 min, the pellets were re-extracted and the supernatants combined. The procedure was repeated for a total number of three times. Further, samples were evaporated using a rotary evaporator (Heidolph Hei-VAP Platinum 3) and then subject to a freeze-drying procedure (VaCo-Series, Zirbus, Germany). Four different samples were obtained as following: NSS water extract (NSS_W), NSS methanol extract (NSS_MeOH), AU water extract (AU_W), and AU methanol extract (AU_MeOH). The freeze-dried samples were quantitatively and qualitatively analyzed as further described.

### 2.4. Total Polyphenol Content (TPC)

The TPC was determined using the Folin−Ciocalteu method as described in Pop et al., (2018). Accordingly, 25 μL of the solubilized freeze-dried sample was mixed with 125 μL (0.2 N) of Folin−Ciocalteu reagent and 100 μL (7.5% *w*/*v*) of sodium carbonate (Na_2_CO_3_) solution. After homogenization, the mixture was incubated for 2 h (room temperature) in the dark [35]. A 96-well plates Synergy HT Multi-Detection Microplate Reader (BioTek Instruments, Inc., Winooski, VT, USA) was used to read the absorbance at 760 nm. The results were interpreted using a calibration curve made with gallic acid (R^2^ = 0.9945) and expressed as gallic acid equivalents (GAE) in mg/g dry weight (DW) of extracts. Triplicate analysis was performed for each extract and the results were presented as mean values ± standard deviations.

### 2.5. Antioxidant Activity Test

The DPPH radical-scavenging activity of AU and NSS extracts was evaluated following Brand-Williams [36] with small modifications. A mixture composed of 250 μL sample and 1750 μL of 0.02 mg/mL 1,1-diphenyl-2-picryl hydrazyl radical in methanol (DPPH solution) was incubated for 30 min at room temperature. The 96-well plates Synergy HT Multi-Detection Microplate Reader (BioTek Instruments, Inc., Winooski, VT, USA) was used to read the absorbance at 517 nm. The calibration curve (R^2^ = 0.9985) was done using Trolox. The control was done with methanol in the same concentration. Triplicate analysis was performed for each extract and the results were presented as mean values ± standard deviations.

### 2.6. FTIR Analysis

The FTIR spectra of AU and NSS methanolic and water extracts were measured using a Shimatzu IR Prestige-21 (FTIR) spectrometer with attenuated total reflectance (ATR) and Zinc Selenide Composite internal reflection accessory. The solubilized freeze-dried extracts were put in a uniform layer directly on the ZnSe ATR crystal. The extracts’ spectra were measured from 4000–650 cm^−1^, for a total number of 64 scans. Water and methanol spectra were taken as background according to the scanned extracts. Between measurements, the ATR crystal was always cleaned with acetone.

### 2.7. Liquid Chromatography-Diode Array Detection–Electro-Spray Ionization Mass Spectrometry (HPLC-DAD-ESI MS)

The HPLC-MS analysis of AU and NSS extracts was performed as described in Pop et al., (2013). Shortly, the detection of the compounds was performed using an HPLC (Agilent 1200) with DAD detection which was coupled to a single quadrupole MS (Agilent 6110). Separation was achieved using an Eclipse XDB C18 column (4.6 × 150 mm, 5 μm particle size) (Agilent Technologies, Santa Clara, CA, USA), at room temperature. The method was performed according to Pop et al., (2013). The mobile phases used were 0.1% acetic acid/acetonitrile (99:1) in distilled water (*v*/*v*) and 0.1% acetic acid in acetonitrile (*v*/*v*) [37]. The eluent flow was 0.5 mL/min. The compounds’ spectra were registered between 200 and 600 nm. The ESI-MS fragmentation source was set in the (+) mode, with a capillary voltage of 3000 V, a temperature of 350 °C, and a nitrogen gas flow of 8 L/min. The molecules were scanned from 100 to 1000 *m*/*z.* Finally, the specific chromatograms for phenolic acids (280 nm) and flavonols (340 nm) were taken into consideration. Agilent ChemStation Software (Rev B.04.02 SP1, Palo Alto, CA, USA) was used for data analysis. The compounds’ UV-visible spectra, retention time, mass spectra information and authentic standards chromatography (when available) were used for compound identification. Compound concentrations were calculated as thymol equivalents (R^2^ = 0.99).

### 2.8. ITEX/GC-MS Qualitative Analysis of Volatile Compounds

The extraction of volatiles from the samples was performed using the in-tube extraction technique (ITEX) followed by their separation and identification by gas-chromatography–mass spectrometry (GC-MS), carried out on a GC-MS Shimadzu model QP-2010 (Shimadzu Scientific Instruments, Kyoto, Japan) equipped with a Combi-PAL AOC-5000 autosampler (CTC Analytics, Zwingen, Switzerland) and a capillary column (ZB-5 ms, 30 m × 0.25 mm i.d. × 0.25 µm, Phenomenex, Torrance, CA, USA).

In brief, for the extraction of the volatiles, the headspace vial containing 0.2 g of powder from AU and NS samples, or 0.5 mL of samples extracts, was closed hermetically and incubated at 60 °C under continuous agitation (500 rpm) for 20 min. After incubation, using the needle of the headspace syringe, an aliquot from the gaseous phase from the vial was repeatedly adsorbed (30 strokes) into a porous polymer fiber microtrap (ITEX-2TRAPTXTA, Tenax TA 80/100 mesh, ea) placed between the syringe needle and body. The volatiles adsorbed into the ITEX fiber were directly injected into the GC-MS injector by thermal desorption. After each analysis, the hot trap (250 °C) was cleaned with N_2_. All of the above operations were perform automatically using an equipment autosampler (Combi-PAL AOC-5000 autosampler, CTC Analytics, Zwingen, Switzerland).

The separation of volatile compounds on the ZB-5ms capillary column was performed using the method described in Pop et al., 2020. Shortly, the column temperature program started from 50 °C (held for 2 min) and increased to 160 °C with a rate of 4°/min and then to 250 °C with a 15°/min rate and held at 250 °C for 10 min. The carrier gas was helium (1 mL/min) and the temperature for the injector, ionic source and interface was set at 250 °C. The MS detection was performed on a quadrupole mass spectrometer operating in full scan (40–500 *m*/*z*), with electron impact (EI) as ion source at an ionization energy of 70 eV [38]. The tentative identification of the volatile compounds was achieved by matching their recorded mass spectra and the fragmentation patterns with those from the software’s NIST27 and NIST147 mass spectra libraries (considering a minimum similarity of 85%) and verified with retention indices drawn from www.pherobase.com or www.flavornet.org for columns with a similar stationary phase. Relative peak areas were expressed as arbitrary units (a.u.), to allow the quantitative comparison of the compounds identified in the samples and their extracts. Thus one a.u. was considered to be 100,000 units of peak area.

### 2.9. Cell Culture

H9c2 cells were grown to confluence in Dulbecco’s modified Eagle’s Medium (DMEM; Microgem, Charlottesville, VA, USA) with 10% fetal bovine serum (FBS; Microgem) and antibiotics (25 U/mL penicillin and 25 U/mL streptomycin) under an atmosphere of 95% air/5% CO_2_ at 37 °C. H9c2 cells were seeded in DMEM 10% FBS at right confluence for each experiment and exposed to NSS and AU extracts (50 μg·mL^−1^) for 4 h. Extracts were then washed away and doxo (1 µM) or fresh medium was added for other 20 h.

### 2.10. Viability Assay

Cell viability was performed using a colorimetric assay based on the 3-(4,5-dimethyl-2-thiazyl)-2,5-diphenyl-2*H*-tetrazolium bromide (MTT) reagent. H9c2 rat cardiomyoblasts (5 × 10^4^ cells/well) were plated into 96-well plates and treated as described above. Cell viability was then assessed as previously reported using the MTT assay (Saturnino et al., 2014). Briefly, 25 µL of MTT (5 mg·mL^−1^) was added and the cells were incubated for 3 h. Thereafter, cells were lysed, and the dark blue crystals solubilized with 100 µL of a solution containing 50% (*v*/*v*) *N*,*N*-dimethylformamide, 20% (*w*/*v*) SDS with an adjusted pH of 4.5. The optical density (OD) of each well was measured with a microplate spectrophotometer (TitertekMultiskan MCC/340; Cornaredo, Milan, Italy) equipped with a 620 nm filter. The viability of the cell line in response to treatment with tested compounds was calculated as percent dead cells = 100 − (OD treated/OD control) × 100.

### 2.11. Measurement of Intracellular ROS

Reactive oxygen species (ROS) formation was evaluated using the probe, 2′,7′-dichlorofluorescin-diacetate (H_2_DCF-DA), as previously reported (Saturnino et al., 2017). Briefly, H9c2 cells (3.0 × 10^5^/well) were plated into 24-well plates and treated as described above. H9c2 were then collected, washed twice with PBS and then incubated in PBS containing H_2_DCF-DA (10 µM) at 37 °C. After 45 min, cells fluorescence was evaluated using a fluorescence-activated cell sorting (FACSscan; Becton Dickinson, Franklin Lakes, NJ, USA) and elaborated with Cell Quest software.

### 2.12. Measurement of Mitochondrial Superoxide Evaluation with MitoSOX Red

Mitochondrial superoxide formation was evaluated using MitoSOX Red, a fluorogenic dye for highly selective detection of superoxide in the mitochondria of living cells that, once targeted to the mitochondria, is oxidized by superoxide and emits red fluorescence. Briefly, H9c2 cells (4.0 × 10^5^ cells/well) were plated in 6-well culture plates and treated as described above. After the incubation period, MitoSOX Red (2.5 µM) was added for 15 min at 37 °C before fluorescence evaluation using flow cytofluorometry. Cell fluorescence was evaluated using fluorescence-activated cell sorting and analyzed with CellQuest software.

### 2.13. Measurement Mitochondrial Membrane Depolarization with TMRE

Mitochondrial membrane depolarization was measured using tetramethylrhodamine methyl ester (TMRE), a fluorescent dye that readily accumulates in active mitochondria in inverse proportion to Δψm, according to the Nernst equation due to its positive charge. For these experiments, H9c2 cells (4.0 × 10^5^ cells/well) were seeded in 6-well tissue plates and treated as described above. Cells were then collected, washed twice with phosphate buffer saline (PBS) buffer and then incubated in PBS containing TMRE (5 nM) at 37 °C. After 30 min, cell fluorescence was evaluated using a fluorescence-activated cell sorting and analyzed with CellQuest software (Milan, Italy).

### 2.14. Data Analysis

Data are reported as mean ± standard deviation (S.D.) values of at least three independent experiments. Statistical analysis was performed by analysis of variance test (ANOVA), and multiple comparisons were made by Bonferroni’s test. A *p*-value < 0.05 was considered as significant.

## 3. Results

The quantity of total polyphenolic compounds generally assumed to be responsible for the antioxidant properties of plant extracts, as well as their antioxidant activity, was determined for both NSS and AU extracts (Table 1). Among the two investigated plant extracts, both AU water and methanol extracts had higher TPC compared with NSS and, interestingly, lower antioxidant activity. Thus, the highest TPC value registered for AU methanolic extract was approximately 17% higher than the highest value registered for NSS methanol extract. Overall, for both plants, TPC and DPPH values were higher in the case of methanolic extracts. Thus, in the case of NSS, the methanol extract had with 19.28% higher values than the water extract, while the antioxidant DPPH activity of water extract was 40% lower than the methanolic extract. Different percentages were observed in the case of AU, where the water extract had TPC and antioxidant capacity assays values lower at approximately 9 and 50%, respectively, compared with the methanolic extracts.

### 3.1. FT-IR Analysis

In order to have a better overview of all compounds that could be extracted in the water and methanolic extracts of AU and NSS, the qualitative analysis was further performed using the FT-IR spectroscopy. The identification of the possible extracted compounds was performed based on the different absorption spectra registered according to the different types of chemical bonds or functional groups existing in the extracted compounds. The next figure (Figure 1) presents the comparative general FT-IR (3500–600 cm^−1^) spectra as well as their fingerprint region (1800–800 cm^−1^) of AU and NSS water and methanolic extracts.

*Allium ursinum* had 13 major peaks, while NSS had 10 major peaks. The peaks in the general FTIR spectra, in the range of 3500–3000 cm^−1^, are represented by the O-H bond stretching vibration suggesting the presence of carbohydrates or amino acids. Next, the two intense absorption bands in the range of 3000–2800 cm^−1^ are represented by the C-H stretching vibrations in lipids [35,39]. The fingerprint region of the two extracts indicated the presence of various compounds. Among common compounds identified in both extracts were lipid esters, indicated by the C=O stretching vibration at 1742, 1737, and 1711 cm^−1^ peaks [40]. The absorption bands at 1648, 1603, 1456 cm^−1^ could be assigned in both extracts to C=C aromatic rings stretching vibrations and C=O in the flavonoid structure [41]. Bands at 1371, 1375, 1254 cm^−1^ could be attributed to C-OH deformation vibrations in flavonoid compounds [42]. Other possible interpretations for the AU extract could be that the band absorbing at 1371 cm^−1^ could be assigned to the C=S stretching vibration existing in sulfur compounds [43]. The C-OH stretching vibrations also encountered in the flavonoids were represented in the extracts by the bands absorbing at 1170, 1165, 1089, 1027 cm^−1^ [42]. In the case of AU extracts, the band absorbing at 1089 and 1027 cm^−1^ could be attributed to the characteristic SO3 symmetric stretching vibration indicating the presence of acids, RSO3 ionic sulphones or other sulphone compounds [43]. The presence of major compounds in NSS like thymoquinone, thymol, carvacrol, and p-cymene could be attributed to the C-H out of plane (peak absorbing at 803 cm^−1^), C-H (CH3) (peak absorbing at 1514 cm^−1^), C-H symmetric and asymmetric stretching (peaks at 1380, 1370, 1360 cm^−1^) already mentioned [40,44].

### 3.2. LC-MS Analysis

The individual composition and content of phenolic compounds were identified using LC-MS. The qualitative and quantitative results are summarized in Table 2 and Table 3. In total, 13 compounds were identified in NSS extracts and 12 in AU extracts. The identification was done considering compound retention times, their UV-Vis absorption spectra, their [M + H]^+^ protonated molecules and literature data. The identified compounds in NSS belonged to various classes like phenolic acids (peaks 1, 2, 4 and 5), terpenic phenols (peaks 11 and 13), alkaloids (peaks 6, 7) and flavonoids (peaks 8, 9, 10, 12) (Table 2). The annotated compounds in AU could also be framed to various classes like phenolic acids (peaks 1,2, and 3) and flavonoids (peaks 5, 6, 8, 9, 10, 11 and 12) (Table 3).

### 3.3. ITEX/GC-MS Analysis

The individual composition and content of volatile compounds were identified using GC-MS. The qualitative results are summarized in Table 4 and Table 5. In total, 20 compounds were identified in NSS extracts and 22 in AU extracts.

### 3.4. Effect of NSS and AU Extracts on Cell Viability Assay

In our experimental conditions, doxo administration significantly (*p* < 0.005) reduced H9c2 cell viability. NSS water extract (#1) and methanol extract (#2) administered 4 h before doxo significantly (*p* < 0.001) preserved H9c2 cells viability. Our results showed that AU water extract (#3) pretreatment had no effect on cell viability and AU methanol extract (#4) further increased doxo-induced effects on cell viability (Figure 2).

### 3.5. Effect of NSS and AU Extracts on Doxo-Induced ROS Release

As reported in Figure 3, both water and methanol NSS extract significantly (*p* < 0.005) reduced doxo-induced ROS production. Similar results were observed in H9c2 pretreated with AU water extract, while AU methanol extract didn’t affect doxo-induced ROS production.

### 3.6. Effect of NSS and AU Extracts on Doxo-Induced Mitochondrial ROS Release

As expected, doxo administration significantly (*p* < 0.001) increased mitochondrial ROS production. Both NSS water and methanol extract significantly (*p* < 0.005) reduced doxo-induced mitochondrial ROS overproduction. Regarding AU, water extract significantly (*p* < 0.05) reduced doxo-induced mitochondrial ROS production, while no effects were observed in AU methanol extract pretreated H9c2 cells (Figure 4).

### 3.7. Effect of NSS and AU Extracts on Doxo-Induced Mitochondrial Membrane Depolarization

In previous work (Pecoraro et al., 2015), depolarization of the mitochondrial membrane in doxo-induced H9c2 cells was observed using TMRE, a fluorescent probe capable of crossing only the membrane of healthy mitochondria. Here, our data showed a marked reduction in the percentage of TMRE-positive cells in doxo-treated cells, so neither NSS nor AU extracts were able to reduce doxo-induced mitochondrial membrane depolarization (Figure 5).

## 4. Discussion

The results of the present study showed that both AU and NSS represent important sources of bioactive compounds (mainly flavonols, phenolic acids, volatile and sulfur compounds) with strong in vitro antioxidant activity.

The quantities of total phenolic compounds identified in NSS and AU extracts were in the same range as the previously reported results. For NSS methanolic and water extracts, 2.64 mg GAE/g DW and 1.67 mg GAE/g DW, respectively, were found [53]. The median value of 2.58 mg GAE/g DW identified in our study for the Romanian AU extracts was slightly higher than the one identified in the Polish AU extracts (0.7 mg GAE/g fresh weight) [54] and lower than the one identified in Montenegro and Bosnia (14.02 mg GAE/g DW) and Herzegovina (16.06 mg GAE/g DW), respectively [55]. The DPPH antioxidant capacity was found higher for methanolic extracts compared with the water extracts. Our findings are in agreement with previously reported results [55,56]. It is known that higher TPC content involves higher DPPH antioxidant activities as a result of the degree of hydroxylation in the phenolic compounds [57], as it was in our case when different extracts belonging to the same plant species were compared. If we compared the two plant species, the highest TPC content determined in the AU methanolic extract did not possess the highest antioxidant capacity. This fact can be explained by the different compounds’ compositions, which, in turn, determines their different antioxidant capacities.

The LC-MS analysis was focused on the identification of phenolic compounds, while the FT-IR analysis included the general composition of all bioactive compounds classes from the extracts. Complementary GS-MS analysis aimed to offer the volatile compounds’ profile compositions and further confirm the presence of sulfur containing compounds existing in the water extract, previously identified by the FT-IR analysis. The LC-MS analysis indicated the presence of coumaric, ferulic and sinapic acids and of other seven kaempferol derivatives compounds. Besides sulfur containing compounds, the FT-IR analysis showed the presence of carbohydrates, amino acids, lipids and flavonoids common for both extracts. GS-MS identified the sulfur-containing compounds only in the AU water extracts (allyl monosulfide; disulfide, methyl 2-propenyl; dimethyl trisulfide; trisulfide, methyl 2-propenyl; tetrasulfide, dimethyl). It was demonstrated that these sulfur compounds have antioxidant activity [58,59,60] and this could have also been contributing to an increased antioxidant capacity of the AU water extract.

Next, the in vitro tests showed that all the extracts did not show any toxicity as demonstrated by unaltered cellular viability. Therefore, we tested their effects on general or mitochondrial ROS release and doxo-induced mitochondrial membrane depolarization. We hypothesized that both NSS and AU extracts can interfere with the antioxidant system without affecting the mitochondrial integrity.

The results obtained after AU methanolic extract cellular treatment showed no effect on cellular viability, doxo-induced ROS production, doxo-induced mitochondrial ROS release and on doxo-induced mitochondrial membrane depolarization. The results can be preliminarily linked to AU methanolic chemical composition in which the sulfur compounds were not identified. Even though several studies evaluated AU antioxidant capacity [61], to our knowledge these were not tested before for their protective effect against oxidative doxo induced damage. Furthermore, other studies are needed to understand the mechanism underlying the reduction in cell viability observed in H9c2 cells pretreated with AU methanolic extract and then exposed to doxo.

Concerning NSS, the antioxidant capacity was tested in several models of doxo-induced cardiotoxicity [62,63]. Indeed, previous studies have documented that pretreatment with thymoquinone, the main active constituent in NSS oil, protected organs against oxidative damage induced by a variety of free radical generating agents, including doxo [64]. Furthermore, it has been proved that all the tested compounds from *N. sativa* exerted strong antioxidant effects, since thymol acted as singlet oxygen quencher, while thymoquinone and dithymoquinone showed superoxide dismutase (SOD)-like activity [65]. *N. sativa* hydro-alcoholic extracts were able to decrease the oxidative marker MDA but increase antioxidants, including SOD, and total thiol concentrations in rat Adriamicyn-induced kidney damage [65].

It is known that doxo is a widely spread chemotherapeutic drug used in the treatment of several illnesses like lymphomas, acute leukemia or various types of solid tumor, but with important health complications that may also lead to multidrug resistance [66]. The health complications usually resulted from doxo-induced cardiotoxicity. Taking into account that the induced cardiotoxicity involves several mechanisms, like increased reactive oxygen species (ROS) production, increased lipid peroxidation, mitochondrial function deterioration and calcium overloading [67] it is very important to find viable pathways to reduce or stop these interconnected mechanisms. Among the identified doxo-induced cardiotoxicity mechanisms, increased ROS plays a central role in doxo cardiotoxic effects development [67], acting as an important trigger to cell death including apoptosis, necrosis and autophagy [68].

In this regard, our study demonstrated that all extracts (excepting the AU methanolic extract) had a positive effect in preventing the production of general or cardiomyocyte ROS production, being involved in the associated mechanisms, and had no effect on mitochondrial membrane depolarization. Accordingly, these results suggest that NSS methanolic and water, and AU water extracts influenced cellular metabolism and viability (explained by the decreased ROS production and the increased cellular viability), and had no influence on cellular electrical activity. This observation is sustained by the negative results obtained on membrane depolarization after pretreatment with investigated plant extracts. Ion channels represent an important target in obtaining cardioprotective effects, especially through vascular tone and cardiac electric activity regulation [69], thus preventing possible malignant arrhythmias. During therapeutic drug development, one of the reasons why investigated compounds are frequently discarded is because of their potential risk in inducing cardiac arrhythmias as a possible result of K^+^ channel inhibition or Na^+^ channels incomplete inactivation [70]. Given that NSS and AU extracts did not affect the cellular electrical activity, we can assume that they can be considered as promising candidates in reducing induced doxo cardiotoxicity by improving cardiac cell metabolism rather than influencing electrical activity. It might be interesting to investigate the effect of NSS and AU extracts on electrical activity in myocytes and specialized tissue, especially on membrane Na^+^ and Ca^2+^ channels.

The main strength of the study is that it presents the effects of two natural extracts rich in active pharmacological compounds on doxorubicin cardiac toxicity, one of its most significant adverse effects observed in clinical practice. Still, the present study has some limitations. First, the results reflected the influence of these extract on mitochondrial activity, but the results were not correlated with their effect on cell membrane electrical activity. Second, the research needs confirmation in in vivo studies on animals to investigate the exact changes in heart electrical activity.

## 5. Conclusions

*Nigella sativa* and *Allium ursinum* represent important sources of bioactive compounds with increased in vitro antioxidant DPPH activities and require different extraction solvents to obtain the targeted pharmacological effects. While both methanolic and water extracts of NSS reduced general and mitochondrial ROS release in the doxorubicin-induced cardiotoxicity cellular model, in the case of AU, only the water extract reduced ROS production. The extracts did not affect membrane depolarization, suggesting that they don’t influence cellular electrical activity. Further studies are needed to verify the effects of these extracts on other pathways (e.g., mitochondrial calcium homeostasis) involved in doxo-induced cardiotoxicity. Detailed pharmacological studies of NSS and AU extract on cardiovascular ion channels, on ion currents and tissue preparations need to be further addressed.

## Figures and Tables

**Figure 1 molecules-25-05259-f001:**
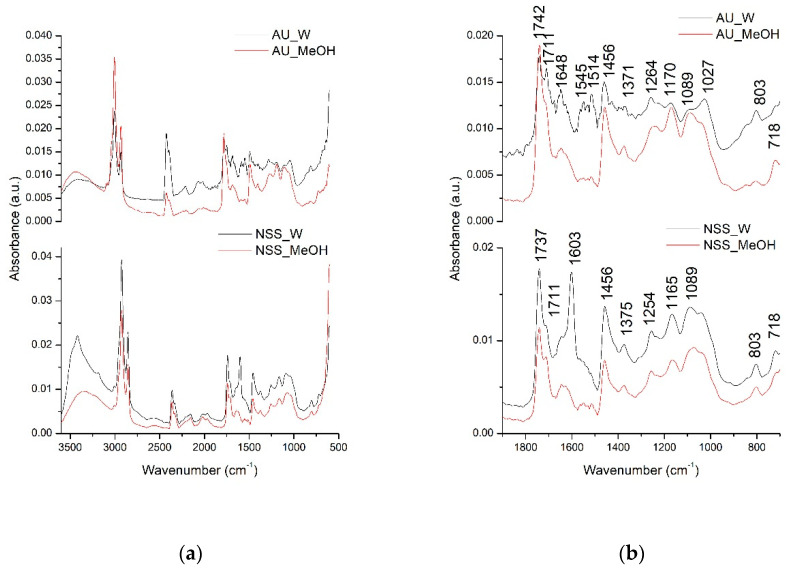
*Allium ursinum* (A.U)- and *Nigella sativa* seed (NSS) general FTIR spectra (500–3500 cm^−1^) (**a**) and fingerprint region (800–1800 cm^−1^) (**b**).

**Figure 2 molecules-25-05259-f002:**
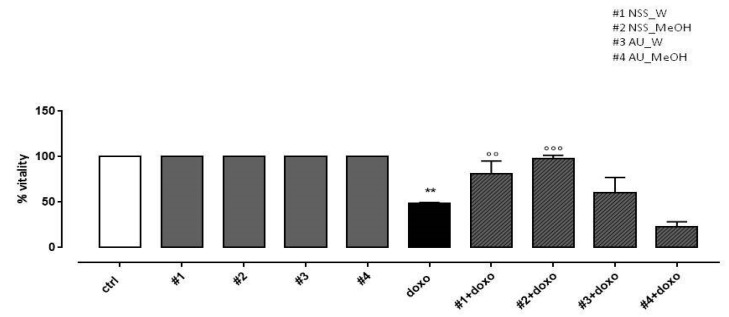
Effects of NSS and AU extracts on H9c2 cell viability. NSS and AU extracts (50 µg/mL) were administered alone for 24 h or 4 h before doxo (1 µM). Cellular viability was assessed by MTT assay. Data are reported as mean ± SD of three independent experiments (N = 6). Statistical analysis was performed by analysis of variance test (ANOVA), and multiple comparisons were made by Bonferroni’s test. ** *p* < 0.005 vs. ctrl; °° *p* < 0.005 and °°° *p* < 0.001 vs. doxo.

**Figure 3 molecules-25-05259-f003:**
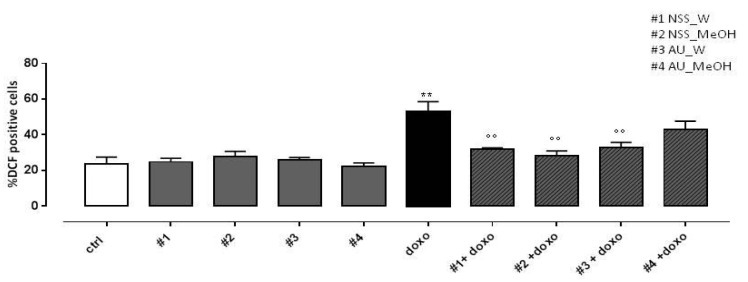
Effect of NSS and AU extracts on doxo-induced ROS release. NSS and AU extracts (50 µg/mL) were administered alone for 24 h or 4 h before doxo (1 µM). Reactive oxygen species (ROS) formation was evaluated using the probe dichlorofluorescein diacetate (H_2_DCF-DA) in H9c2 cells. ROS production was expressed as the mean ± SD of the percentage of DCF positive cells of at least three independent experiments (N = 6). Statistical analysis was performed by analysis of variance test (ANOVA), and multiple comparisons were made by Bonferroni’s test. ** *p* < 0.005 vs. ctrl; °° *p* < 0.005 and vs. doxo.

**Figure 4 molecules-25-05259-f004:**
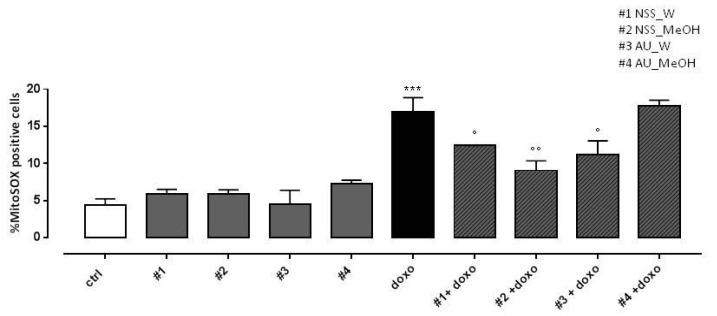
Effect of NSS and AU extracts on doxo-induced mitochondrial ROS release. NSS and AU extracts (50 µg/mL) were administered alone for 24 h or 4 h before doxo (1 µM). ROS formation was evaluated using the probe MitoSOX red in H9c2 cells. Mitochondrial superoxide production was expressed as mean ± SD of the percentage of MitoSOX positive cells of at least three independent experiments (N = 6). Statistical analysis was performed by analysis of variance test (ANOVA), and multiple comparisons were made by Bonferroni’s test. *** *p* < 0.001 vs. ctrl; ° *p* < 0.05 and °° *p* < 0.005 and vs. doxo.

**Figure 5 molecules-25-05259-f005:**
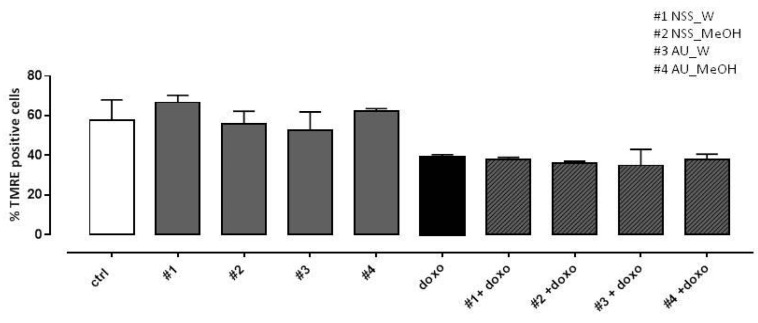
Effect of NSS and AU extracts on doxo-induced mitochondrial membrane depolarization. NSS and AU extracts (50 µg/mL) were administered alone for 24 h or 4 h before doxo (1 µM). The mitochondrial membrane potential was evaluated by flow cytometry analysis with Tetramethylrhodamine ethyl ester (TMRE), a cell-permeant, positively-charged, red-orange dye which penetrates and accumulates in the mitochondria in inverse proportion to the membrane potential. The low value of percentage of TMRE^+^ cells means that the TMRE dye was not trapped in the mitochondrial membrane due to its depolarization. Data were expressed as mean ± SD of fluorescence intensity of at least three independent experiments (N = 6).

**Table 1 molecules-25-05259-t001:** Total phenolic content and antioxidant activity of NSS and AU samples.

Samples	TPC (mg GAE/g DW)	DPPH (mM TE/g DW)
NSS_W	1.80 ± 0.06	297.30 ± 1.27
NSS_MeOH	2.23 ± 0.13	488.37 ± 1.20
AU_W	2.46 ± 0.12	158.47 ± 1.55
AU_MeOH	2.70 ± 0.08	312.15 ± 1.26

Abbreviations: TPC—Total Polyphenol Content; GAE—gallic acid equivalents; DPPH—2,2-diphenyl-1-picryl-hydrazyl-hydrate; DW—dry weight; T—Trolox equivalents per gram of dry plant material.

**Table 2 molecules-25-05259-t002:** Tentative identification, characterization and concentration of major compounds identified in *Nigella sativa* seeds water and methanolic extracts.

No	Time	M/[M + H]^+^	λ_max_ (nm)	Tentative Identification	Concentration mg/g Dry Weight (Thymol Equivalents)		References
					NSS_W	NSS_MeOH	
1	2.95	224	275	Sinapic acid	14.28 ± 0.09	8.56 ± 0.07	[45]
2	4.07	138	270	Hydroxybenzoic acid	4.17 ± 0.05	2.43 ± 0.06	[46]
3	7.87	792	290	Unknown	3.88 ± 0.05	2.54 ± 0.06	-
4	10.23	432	298	Benzyl alcohol dihexoside	6.38 ± 0.04	5.12 ± 0.06	[47]
5	12.66	198	278, 320	Syringic acid	5.21 ± 0.08	4.78 ± 0.11	[46]
6	13.5	196	295	Damascenine	2.18 ± 0.06	2.24 ± 0.07	[47]
7	13.95	342	278, 300	Norargemonine	3.61 ± 0.06	8.89 ± 0.05	[47]
8	14.27	772	270, 350	Quercetin-rhamnosyl-diglucopyranoside	4.76 ± 0.03	6.00 ± 0.04	[47]
9	16.39	594	265, 335	Kaempferol-rhamnosyl-diglucoside	0.56 ± 0.04	0.82 ± 0.05	[47]
10	17.14	755	265, 355	Kaempferol-rhamnosyl- Diglucopyranoside	1.57 ± 0.06	3.27 ± 0.05	[47]
11	21.21	474	260, 300	Thymol-sophoroside	2.13 ± 0.05	1.62 ± 0.10	[47]
12	23.25	286	260, 353	Kaempferol	1.60 ± 0.05	4.96 ± 0.13	[47]
13	24.98	150	260	Thymol	1.36 ± 0.07	5.09 ± 0.06	[47]

**Table 3 molecules-25-05259-t003:** Tentative identification, characterization and concentration of major compounds identified in *Allium ursinum* water and methanolic extracts.

No	Time	M/[M + H]^+^	λ_max_ (nm)	Tentative Identification	Concentration mg/g Dry Weight (Thymol Equivalents)		References
					AU_W	AU_MeOH	
1	2.99	164	217, 280	Coumaric acid	44.49 ± 0.02	39.64 ± 0.07	[48]
2	3.11	194	218, 295	Ferulic acid	11.97 ± 0.03	16.08 ± 0.04	[48]
3	3.19	224	238, 326	Sinapic acid	12.53 ± 0.03	13.78 ± 0.03	[48]
4	4.06	298	270	Unknown	17.26 ± 0.03	18.68 ± 0.06	-
5	11.37	756	265, 355	Kaempferol-glucosyl-rhamnosyl-glucoside	4.17 ± 0.04	9.03 ± 0.03	-
6	12.67	448	264, 350	Kaempferol-glucopyranoside	8.34 ± 0.04	6.48 ± 0.04	[49]
7	14.63	540	230, 267, 320	Unknown	9.97 ± 0.02	20.87 ± 0.05	-
8	15.73	490	260, 330	Kaempferol-acetylglucoside	6.02 ± 0.04	12.78 ± 0.05	[50]
9	16.14	903	265, 315	Kaempferol-deoxyhexose-hexoside -p-coumaroyl hexoside derivative	6.15 ± 0.04	15.46 ± 0.02	[50]
10	17.24	448	267, 362	Kaemferol-glucoside	9.98 ± 0.02	9.16 ± 0.09	[51]
11	20.81	491	266, 330	Kaempferol glucosyl-acetate	1.31 ± 0.04	3.25 ± 0.07	[51]
12	23.30	286	265, 365	Kaempferol	2.39 ± 0.03	4.86 ± 0.07	[52]

**Table 4 molecules-25-05259-t004:** Tentative identification of major compounds identified in *Nigella sativa* seeds comparative with *Nigella sativa* seeds water and methanolic extracts.

No	Time	Tentative Identification	Concentration (Arbitrary Units—a.u.)		
			NSS	NSS_W	NSS_MeOH
1	7.65	α-Thujene	954.94	803.33	41.21
2	7.89	α-Pinene	178.19	150.59	2.09
3	8.25	Bicyclo [3.1.0]hex-2-ene, 4-methylene-1-(1-methylethyl)	1.50	-	-
4	8.47	Camphene	2.43	-	-
5	9.28	Sabinene	87.79	40.78	7.18
6	9.46	β-Pinene	147.32	88.79	12.67
7	9.88	β-Myrcene	5.34	0.73	-
8	10.43	n.i.		1.35	-
9	10.52	α-Phellandrene	3.83	-	-
10	10.91	α-Terpinene	30.93	13.07	2.68
11	11.24	p-Cymene	1707.59	715.18	289.50
12	11.38	D-Limonene	108.43	57.03	14.79
13	12.49	γ-Terpinene	70.89	28.28	14.77
14	13.54	Terpinolene	6.28	1.38	-
15	13.75	n.i.	0.73	-	-
16	13.94	n.i.	10.36	-	2.40
17	14.82	n.i.	39.07	1.46	12.11
18	17.23	1-Terpinen-4-ol	0.83	-	1.63
19	17.57	1-Undecene	-	-	4–35
20	17.96	n.i.	-	-	2.19
21	19.80	Thymoquinone	124.80	1.49	111.35
22	21.53	Thymol	-	-	8.32
23	23.46	α-Longipinene	1.09	2.12	1.05
24	24.85	1-Dodecanol	-	-	3.50
25	25.55	1,4-Methanoazulene, decahydro-4,8,8-trimethyl-9-methylene-,[1S-(1-α.,3a-β,4-α,8a-β)]-	4.90	8.79	7.77

**Table 5 molecules-25-05259-t005:** Tentative identification of major compounds identified in *Allium ursinum* comparative with *Allium ursinum* water and methanolic extracts.

No	Time	Tentative Identification	Concentration (Arbitrary Units—a.u.)		
			AU	AU_W	AU_MeOH
1	4.15	Hexanal	1.07	-	-
2	5.54	Allyl monosulfide	-	1.02	-
3	7.26	Disulfide, methyl 2-propenyl	-	11.25	-
4	7.64	α-Thujene	5.18	0.85	-
5	7.89	α-Pinene	5.65	1.76	-
6	8.47	Camphene	2.36	-	-
7	8.89	Benzaldehyde	-	0.49	-
8	9.091	Dimethyl trisulfide	-	11.07	-
9	9.28	Sabinene	1.07	-	-
10	9.46	β-Pinene	1.54	-	-
11	9.89	β-Myrcene	3.049	2.59	-
12	11.21	p-Cymene	20.46	4.13	-
13	11.37	d-Limonene	5.81	6.23	-
14	11.50	n.i.	1.09	1.21	-
15	12.76	Acetophenone	-	-	0.87
16	15.53	Trisulfide, methyl 2-propenyl	-	3.49	-
17	16.29	Benzoic Acid	1.30	-	0.13
18	16.78	Benzoic acid, ethyl ester	-	-	1.76
19	17.57	1-Dodecene	-	-	7.52
20	18.49	Tetrasulfide, dimethyl	-	0.84	-
21	24.86	3-Hexadecene, (*Z*)-	-	-	13.24
22	25.56	n.i.	0.55	-	-
23	31.04	1-Pentadecene	-	-	15.33
24	34.08	1-Heptadecene	-	-	2.94

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
