# Peer review of "Evaluation of the Antioxidant Activity of Nigella sativa L. and Allium ursinum Extracts in a Cellular Model of Doxorubicin-Induced Cardiotoxicity"

_molecules, 2020, doi:10.3390/molecules25225259_

Round 1

Reviewer 1 Report

The authors addressed some issues, however, some of them are still open. At first, any improvement was done regarding the phytochemical analysis. As the manuscript is written, it is suitable rather for some journal focused on ethnopharmacology/phytomedicine. However, when deeper phytochemical analysis (identification/quantification of more compounds, detection of sulphuric compounds in AU extracts, etc.) will be done, it could be acceptable for Molecules.

Before publishing, authors should also make these revisions:

  1. Line 261-263 - Exctract #3 did not any effect on cell viability and #4 decresed it. It should be corrected and taken into account in the following discussion.
  2. Line 326-329 - The discrepancy between TPC and DPPH antioxidant effect should be discussed in more detail.
  3. Line 340-344 - The effect of AU extracts in the presence of Doxo should be mentioned in this paragraph.

Author Response

The authors addressed some issues, however, some of them are still open. At first, any improvement was done regarding the phytochemical analysis. As the manuscript is written, it is suitable rather for some journal focused on ethnopharmacology/phytomedicine. However, when deeper phytochemical analysis (identification/quantification of more compounds, detection of sulphuric compounds in AU extracts, etc.) will be done, it could be acceptable for Molecules.

Reply:  A deeper phychemical analysis was further performed by GS-MS. The details were added to the text in the materials and methods section (Lines 156-184), results (table 4 and 5, Lines 290-    ) and discussions (Lines 379-386).

Lines

Before publishing, authors should also make these revisions:

  1. Line 261-263 - Exctract #3 did not any effect on cell viability and #4 decresed it. It should be corrected and taken into account in the following discussion.

Reply: the correct sentence has been added. Now the text is in agreement with the data shown in figure 2. We apologize for the error probably due to an oversight in the revision of the previous text.

  1. Line 326-329 - The discrepancy between TPC and DPPH antioxidant effect should be discussed in more detail.

Reply: Discussed. Line 371-386.

  1. Line 340-344 - The effect of AU extracts in the presence of Doxo should be mentioned in this paragraph.

Reply: As suggested, the effect of AU extracts in the presence of doxo has been mentioned. Our data does not allow us to hypothesize a possible mechanism behind these effects. For this reason, the sentence “other studies are needed to understand the mechanism underlying the reduction in cell viability observed in H9c2 cells pre-treated with AU methanolic extract and then exposed to doxo.” has been added.

Reviewer 2 Report

Authors in this resubmitted revisited paper meet majority of my original comments and I thank for their answers. However, authors did not answer to all my points. Therefore, some revisions are still needed before acceptance of this manuscript.

I have to insist on my point regarding H9c2 methodology - did the co-incubation experiments performed using serum-containing medium? It will provide relevant information for readers, as it will lead to artificially increased viability in controls.

While re-reading the manuscript, I realized another comment. Authors should provide information about the own toxicity of used extracts. In Figure 2 it is expressed as 100 % but not compared to control. Furthermore, there are confusing colors used in the Fig. 2. 

Author Response

Authors in this resubmitted revisited paper meet majority of my original comments and I thank for their answers. However, authors did not answer to all my points. Therefore, some revisions are still needed before acceptance of this manuscript.

I have to insist on my point regarding H9c2 methodology - did the co-incubation experiments performed using serum-containing medium? It will provide relevant information for readers, as it will lead to artificially increased viability in controls.

Reply: The need to use starvation is a controversial topic. If, on the one hand, this allows us not to alter cellular vitality, on the other it can cause cellular stress. All the cells used in our experiments (both control and treated cells) are growth and incubated in complete medium containing 10% fetal bovine serum (FBS). To better clarify, in cell culture section we specified that all experiments were performed in DMEM 10% FBS.

While re-reading the manuscript, I realized another comment. Authors should provide information about the own toxicity of used extracts. In Figure 2 it is expressed as 100 % but not compared to control. Furthermore, there are confusing colors used in the Fig. 2. 

Reply: In figure 2 are reported data on cell viability, that is 100% in control cells. Also for cells treated with our extracts, cell viability was 100% (compared with control), since extracts alone were not cytotoxic. Figure 2 has been corrected and colors are the same of other figures.

Reviewer 3 Report

In this manuscript, Pop RM et.al studied the composition of water and methanolic extracts of AU and NSS, and using a doxo-treated H9c2 cell line as a celllular model of cardiotoxicity, they investigated NSS and AU extracts on doxo-induced cell viability, ROS release and mitochondrial ROS release. The manuscript is well written, but the experimental design and methodology must be improved. As to the experimental design, the authors designed two parts of work, the extracts’ composition and their antioxidant activity, while there’s a gap between these two parts. This work should answer the questions: 1, Do these extracts have antioxidant activity? 2, Which composition is effective in the antioxidant activity? The authors answered the first question but not the second one, although they studied the phenolic composition. It would be better if they further used the phenolic composition (although it is a mixture) for antioxidant activity study. As to the methodological concern in the cellular experiments, after H9c2 cells treated with the extracts for 4 hours and before addition of doxo, how did the authors treat the cells? Did the extracts were washed away completely? Or the mixture may react with doxo and weaken its cytotoxicity. If so, the results in this manuscript was not robust enough to answer the first question above. If the extracts were washed away completely after 4-hour treatment, then the extracts(#1, 2, 3, or 4) +doxo groups should compare with extracts(#1, 2, 3, or 4) which treated for 4 hours + doxo solvent for 20 hours, but not with extracts(#1, 2, 3, or 4) which treated for 24 hours. Moreover, how were these extracts dissolved and what was the solvent controls? Therefore, there’s much room for this manuscript to improve. It is not acceptable for publication in current form.

Author Response

In this manuscript, Pop RM et.al studied the composition of water and methanolic extracts of AU and NSS, and using a doxo-treated H9c2 cell line as a celllular model of cardiotoxicity, they investigated NSS and AU extracts on doxo-induced cell viability, ROS release and mitochondrial ROS release. The manuscript is well written, but the experimental design and methodology must be improved. As to the experimental design, the authors designed two parts of work, the extracts’ composition and their antioxidant activity, while there’s a gap between these two parts. This work should answer the questions: 1, Do these extracts have antioxidant activity? 2, Which composition is effective in the antioxidant activity? The authors answered the first question but not the second one, although they studied the phenolic composition. It would be better if they further used the phenolic composition (although it is a mixture) for antioxidant activity study. As to the methodological concern in the cellular experiments, after H9c2 cells treated with the extracts for 4 hours and before addition of doxo, how did the authors treat the cells? Did the extracts were washed away completely? Or the mixture may react with doxo and weaken its cytotoxicity. If so, the results in this manuscript was not robust enough to answer the first question above. If the extracts were washed away completely after 4-hour treatment, then the extracts(#1, 2, 3, or 4) +doxo groups should compare with extracts(#1, 2, 3, or 4) which treated for 4 hours + doxo solvent for 20 hours, but not with extracts(#1, 2, 3, or 4) which treated for 24 hours. Moreover, how were these extracts dissolved and what was the solvent controls? Therefore, there’s much room for this manuscript to improve. It is not acceptable for publication in current form.

Reply: The extracts have antioxidant activity, since, as reported in figure 3 and 4, they reduced doxo-induced cytosolic and mitochondrial ROS production. This is a preliminary study, so other studies are needed to verify which compounds or mixture of compounds is effective in the antioxidant activity. One possible explication was added.

In extracts- treated cells, H9c2 were treated with the extracts for 4 hours, then the extracts were washed away completely and fresh medium was added for 20 hours. In extracts and doxo co-treated cells, H9c2 cells were treated with extracts for 4 hours, then the extracts were washed away completely and doxo was added in fresh medium. Doxo+extracts groups were compared with doxo alone, since we aimed to highlight the protective effects of extracts in doxo-induced toxicity. To better clarify, this information has been added in the text.

The extracts were dissolved in DMSO, then serial dilutions in DMEM 10% FBS were made. therefore the final dilutions used in the experiments had a DMSO concentration lower than 0.0001. This is the reason why the solvent control was not reported.

Round 2

Reviewer 1 Report

The manuscript has been significantly improved after the 2nd revision. I just recomment to the authors to correct minor mistypes in the text, e.g. exchange decimal comma to decimal full stop in tables, write the number of references in the line 75 instead of the author name (Vlase et al., 2013) etc.

After these corrections the manuscript is sufficient for publication.

Author Response

I just recomment to the authors to correct minor mistypes in the text, e.g. exchange decimal comma to decimal full stop in tables,

Reply: Decimal comma have been exchanged to decimal full stop in all tables

Write the number of references in the line 75 instead of the author name (Vlase et al., 2013) etc.

Reply: In the line 75 the number of reference [27] now replaces the author name.

This manuscript is a resubmission of an earlier submission. The following is a list of the peer review reports and author responses from that submission.

Round 1

Reviewer 1 Report

The manuscript from Pop RM et al. characterizes composition of water and methanolic extracts of wild garlic and black cumin, describes their antioxidant and ROS scavenger properties, and further describes their effect on doxorubicin-treated H9c2 cell line. The manuscript is generally well written, but it has many pitfalls, methodological as well as conceptual, that lower its overall merit and value. Therefore, it is not acceptable for publication in this form.

In abstract, it would be better to place common name of plants together with the Latin ones, and further to use it uniformly, esp. for abbreviation "NS" vs "NSS". The abstract is completely missing the conclusion of the work, and esp. of the results describing relation between decrease of ROS production and antioxidant effects.

intro - based mainly on self-citations. Authors should use more appropriate, up-to-date, and primary references.

Authors cited Henriksen's review paper [3], in which the interaction of ANT with topoisomerase II beta is pointed as a main mechanism of ANT cardiotoxicity, and the only cardioprotective drug, dexrazoxane, possess its effects due to preventing interaction of ANT with TOP2B. This aspect is not mentioned or discussed further. The other citation to ANT cardiotoxicity is self-citations [4,26]. In this regard, the rationale and hypothesis of this work is weak and biased. 

2nd paragraph of intro - these statements are biased and without appropriate explanation of both sides of the issues. Furthermore, they are based on self-citation and non-peer reviewed sources. Nutraceuticals are shown as the best and the most safe alternative to standard CVD treatment, i.e., authorities-approved and verify by RCT. Dealing such a big clinical issue as ANT cardiotoxicity is, it would be worthy to show the effectiveness of plant extracts (esp. studied ones) in RCT, to distinguish incomparable characteristics between in vitro models, animal models etc.

As described in Methods, H9c2 cells were exposed to studied drug in FBS containing DMEM, and thus were in proliferative phenotype. Therefore, effects of doxorubicin observed in this experimental condition is "antiproliferative" effect rather than "cardiotoxic". Furthermore, this model reflects acute toxicity, not chronic. This is why at least all antioxidative compounds failed in chronic animal models of ANT cardiotoxicity or RCT.

Furthermore, H9c2 cell line is known not to express specific cardiomyocyte-related proteins like cTnT of MHC, although this cell line belongs to the mostly utilized model in cardiovascular experiments. Observed effects on mitochondria and cell viability (general markers of toxicity) should be therefore assigned to cytotoxic effect rather that cardiotoxic.

For proper description of extracts' antioxidant effects, ROS scavenger effects, cellular toxicity and protection against DOXO, it would be worthy to show the dose-dependency. What was the rationale for the selected dose? Furthermore, antioxidant capacity is not solely dependent on TPC. Authors should explain discrepancies of compound identification between FTIR and LCMS.

English language and style is generally good, but there are some typos: abbreviation NS vs NSS, DPPH units in Table 1, use an uniform decimal number style.

For each graph, denote the exact "N". use of S.D. instead of S.E.M is more appropriate to describe data distribution. 

Reviewer 2 Report

The present manuscript is a piece of conscientious scientific work. However, according to my opinion, the phytochemical part is very preliminary for publishing in Natural Products Chemistry section of Molecules. I recommend authors to make deeper phytochemical analysis with emphasis on potentialy antioxidant compounds.

Some minor revisions should be also addressed before publishing:

  1. The abbreviation of N. sativa should be uniform during the manuscript - NS vs. NSS.
  2. Line 89-90 - There should be mentioned, who did identify/verify the used plant material and if the specimen is available.
  3. Line 188-198 - The link to Tab. 1 is missing in the text.
  4. Figure 5 - The description of #4 is missing in the graph legend.

Reviewer 3 Report

1 #4 is missing _MeOH in Fig 5.

2 Fig3 and fig5 need to be supplemented with the original image.

3 In P8, L246,“In vitro studies” should be removed.

4 In figure 2,Why is the cell survival rate so low for the treatment of 4 # 4 + D o x o?

5 In P11, L344-345 “in our case, the highest TPC content determined in AU methanolic extract did not possess the highest antioxidant capacity.”The author should explain the possible reasons for this result.

6 In P11,L346-347,“This can be explained by the fact that the level of antioxidants is influenced, besides the species and botanical family by the solvents used in the extraction and by the extraction methods as well [49].”The above sentences, and there is no scientific answer to this question.

7 In P11,L351-353,The FT-IR analysis has shown the presence of sulfur-containing compounds in higher quantities in the water extract than in the methanolic one. The LC-MS analysis did not indicate the presence of sulfur-containing compounds in any extract. Sulfur-containing compounds have antioxidant activity?

8 In P11,L363-364,“Even though several studies evaluated AU
antioxidant capacity [50], to our knowledge, these were not tested before for its protective effect against oxidative doxo induced damage.” The author should study its mechanism in depth.

9 The original result of fig2-5 is the image, so the image should be provided.